# Effectiveness of Reinforcing Bent Non-Uniform Pre-Stressed Glulam Beams with Basalt Fibre Reinforced Polymers Rods

**DOI:** 10.3390/ma12193141

**Published:** 2019-09-26

**Authors:** Agnieszka Wdowiak, Janusz Brol

**Affiliations:** 1Faculty of Civil Engineering and Architecture, Kielce University of Technology, 25-314 Kielce, Poland; 2Faculty of Civil Engineering, Silesian University of Technology, 44-100 Gliwice, Poland; janusz.brol@polsl.pl

**Keywords:** wood, basalt fibre, polymer fibre, mechanical testing, wooden structures’ reinforcement

## Abstract

The article presents the testing designating the impact of structural non-uniformity on the effectiveness of reinforcing bent wooden beams reinforced with basalt fibre (BFRP—Basalt Fibre Reinforced Polymers) rods. The obtained results demonstrate a positive impact of the strengthening in improving the bearing capacity and rigidness of the wooden beams. The article presents the impact of selected physical and chemical properties of wooden elements on the achieved strengthening reliability, increase in bearing capacity and the estimation of the reduction of deflections and stresses of bent beams, made from various wood quality classes and reinforced using BFRP rods. The conducted testing featured an analysis of the ability of using lower quality class lumber to strengthen the beams with pre-stressed basalt fibre rods. This solution allows for reducing the cross-section or lower the class of used wood with simultaneous maintenance of comparable rigidity and bending strength of beams, as in the non-strengthened beams.

## 1. Introduction

The development and promotion of sustainable materials in the civil engineering industry is becoming increasingly important in recent years [1]. Glulam wood, as an engineering product, was developed in the 19th century in Europe, and is currently widely used in buildings and bridges [2,3]. The glulam beams are durable and dimensionally stable, despite their durability parameters sometimes being unsatisfactory because of natural flaws, e.g. knots [4,5]. The glulam beams have a relatively low rigidity, which is why there were many attempts at strengthening them using high tensile strength materials [6]. The increased effectiveness of drying, sorting, gluing and linking lamellas using vee joints contributed to the attainment of a structural solution in the second half of the 20th century that is still used today. This technology is still being developed, and this was also affected by the introduction of innovative glues from the group of melamine-urea-formaldehyde (MUF), polyurethane (PUR) and epoxy resins at the beginning of the 21st century [7,8].

The wooden beams made in the past did not have high durability, which is why they were replaced or strengthened by the traditional methods. Nevertheless, in some cases, their execution was impossible or involved the necessity to completely replace the wooden element. Because of the above reason, there was great interest in the introduction of composite materials, as reinforcements of wooden elements subjected to bending [9,10] or shearing stresses [10,11]. This resulted, among others, from the increase in stress, decay of the element’s mechanical properties, structural imperfections and increases in excessive displacements [10,12]. Fibre-reinforced polymers (FRP) constitute promising materials that can provide many benefits in strengthening and rehabilitating the wooden elements [13,14].

The fibrous composites are characterised by an excellent ratio of strength and weight comparing to other civil engineering materials. The introduction of fibrous composites in the strengthening of structural elements in the infrastructure is effective and presented in several studies [15,16,17,18,19]. The composites are also characterised by high fatigue properties comparing to common structural materials [19,20]. Their use contributes to the reduction of maintenance costs. 

The fibrous composites are very useful as reinforcement glued to wood side surfaces or into its cross-section, when compared to steel, because of steel’s susceptibility to corrosion after contact with the wood’s moisture. Basalt and glass fibre reinforcements seem to be most suitable for wooden elements, which is related to their low price and advantageous mechanical properties [19]. To obtain lower beam height and greater reliability, when using lower quality wood, many methods were developed to improve the static working during glulam and solid wood bending by, among others, placing a metallic reinforcement inside its cross-section. 

A new, increasingly popular solution is the use of fibrous composites (FRP), as wooden beam reinforcement, in the form of sheets, plates, tapes and rods. The focus was also on reinforcing wood by using BFRP basalt rods, because they are derived from natural materials and are economically essential because of their lower price, when compared to CFRP carbon fibre [21,22,23,24]. 

Recently, natural fibres also started to be used as wooden structure reinforcement [23,25,26,27,28]. Certain experimental works were conducted to reinforce wooden beams using pre-stressed steel or FRP materials [23,29,30,31,32,33,34,35,36]. The studies show that the wood’s tensile strength was significantly improved because of the high degree of reinforcement use and full utilisation of the wooden elements’ bearing capacity. Nearly all existing studies demonstrated a positive impact of reinforcing materials on the static working of bent wooden beams [23]. The literature features a limited number of studies concerning low quality glulam wood reinforcement using internal strengthening [19].

The BFRP rods are characterised by good mechanical properties. These materials are light and have good chemical corrosive resistance [16,22]. This solution can be deemed as environment-friendly, because it enables complete recycling without a negative impact on the natural environment and humans [8,37]. Basalt is the most common rock, whereas basalt fibres have a substantially lower thermal activity than steel and other synthetic fibres. In wood engineering, the number of studies in the field of using basalt fibre reinforcement is small. These studies utilised basalt fibres to reinforce low quality glulam wood. It was ascertained that the use of basalt fibre rods is a very effective technology of repairing damaged wooden elements [22]. In addition, pre-stressed basalt fabric was used in the research to determine the wood reinforcement effectiveness [22,38]. The results demonstrated that among all the tested systems, the BFRP fabric and PUR glue were the most suitable [38].

Furthermore, many other researchers conducted theoretical and numerical analyses based on the experimental results from the non-linear workings of reinforced elements [23,30,39,40,41,42,43,44]. Despite this fact, it was still necessary to conduct testing to better meet the requirements concerning production and designing of reinforced wooden structures [23]. As a result, testing of the reinforcement of non-uniform pre-stressed glulam beams with BFRP rods was conducted, and its results are presented below. The following study is based on the results of research carried out during the preparation of the doctoral dissertation of one of the authors of the article [45].

## 2. Testing Materials 

The testing materials used for glulam beams consisting of various configurations of wood quality classes (KS—medium quality class, KG—lower quality class), reinforced with pre-stressed BFRP rods, are specified in Table 1. Each series included three full-size research models (unreinforced—type NWa, reinforced—types Wa-A and Wa-B).

### 2.1. Wood

The experimental testing program covered *Pinus sylvestris* L., derived from the Małopolskie Polish Natural and Forest Lands, and collected at the beginning (April) and end (August) of the growing season [45]. 

The structural sawn wood was segregated in terms of dimensions and origin, with consideration of early and late wood. Then, a humidity measurement was conducted using a GANN hygrometer. The elements were labelled and sorted into parcels, and then transported to drying rooms to achieve the average humidity of 12%. After achieving the adopted parameters, a visual humidity analysis of each piece of sawn wood was conducted according to PN-EN 13183-2:2004 [46] and PN-EN 13183-3:2007 [47]. 

The dried materials were sorted visually by thoroughly inspecting all elements, taking into consideration the layout of flaws, according to PN-D-94021:2013-10 [48]. The existing structural and geometrical features of the structural sawn wood, such as knots, spiral grain, cracks, resin blisters, bark pockets, catfaces, blue stains, rotting, larval gallery, reaction wood, average growth ring width, density, wane, longitudinal curvatures of sides, longitudinal curvatures of planes, width-related lateral curvatures, width-related knottiness were thoroughly counted, with special attention being paid to the knots [48,49,50,51]. The quantity, health, dimensions and location of knots constituted the basis for the wood’s qualitative classification [45]. Based on this analysis, according to PN-D-94021:2013-10 [48], coniferous structural sawn wood intended for glulam beams was divided into three quality classes: KW—exclusive class, KS—medium quality class, KG—lower quality class, and designated for “rejection” [4,5,45,52,53,54,55,56]. Furthermore, the measurements covered the growth ring occurrence and the share of late wood [5,45,57], which is decisive in terms of the increase in specific weight and strength parameters, as well as moisture content, mass and density of wooden samples in a dry air condition. The growth ring width constituted the distance between two outer borderlines of late wood in the trunk’s transverse cross-section radius. The width of the growth rings was determined in mm or in the number of annual growth rings per 1 cm of radius. The average growth ring width was designated by dividing the arithmetic mean of the greatest and the smallest diameter by double the number of annual growth rings. The share of late wood was designated based on linear measurements with simultaneous estimation of the average growth ring width. The measurements were conducted along the medium radius in the sample’s transverse cross-section to determine the differences between the tested technical feature and the share of late wood. The samples were cut out perpendicularly to the axis. A diagonal cutting would cause an increase in the width of the measured zones, which will result in an erroneous specification of the growth ring occurrence. The cross-section’s surface was smoothed out, and then the measurement line, with a perpendicular course, in relation to the growth rings was determined. This line was used to make cuts in the wood with a sharp tool to intensify the growth rings’ articulation [5]. A magnifying glass with a millimetre scale was used for the testing. The scale’s zero point was set on the line dividing the early and late wood, while the scale’s borderline course was between the late wood and the next annual growth ring. The sum of the conducted readings specified the total width of the late wood zones [5,45,57].

The beams made from the elements with various wood quality configuration (see Figure 2 in Section 3) were characterised by structural and geometrical features that were catalogued separately for each testing model. An example of the testing elements’ description is provided in Table 2 for one non-reinforced and one reinforced beam. The location of the extensometer bases on individual lamellas is shown in Figure 4 in Section 3.

### 2.2. Applied Glues

The experience gained during the conducted preliminary testing in preparing the testing elements shows that the prevention of delamination in glue connections between the glulam beam lamellas required the use of liquid glues with substantial infiltration into the wood pores [45]. Figure 1 presents the microstructure of the connection of epoxy resin filling the cellular walls of tracheids in early and late wood in the cross-section with clear borderline of annual increments. The radial cross-section of the tracheid with funnel-shaped lacunae is arranged in parallel, which strengthens the connection’s adhesion. This is important in the scheme of junction destruction and the obtained shear bearing capacities. It can be added that the “wood–glue–rod” composition is caused by the non-uniform structure of wood, rod and the properties of the applied glues and by the transfer of shear forces to the beams’ microstructural elements [45].

The specific wood lamellas were glued with one another using the polyvinyl acetate glue D4 (density of approx. 1.10 g/cm^3^, viscosity 13,000 mPa.s) or LG 385 + HG 385 epoxy glue with liquid consistency. The lamellas were placed so that the core was directed towards the beam’s top part according to PN-EN 14080:2013-07 [58]. 

The LG 385 + HG 385 epoxy resin was used for the “glue–rod” connection. The resin-based adhesive layer was achieved as result of mixing the LG 385 epoxy resin (density of 1.18 ÷ 1.23 g/cm^3^, viscosity 600 ÷ 900 mPa.s), with the HG 385 hardener (density of 0.94 g/cm^3^, viscosity 50 ÷ 100 mPa.s). After mixing the resin and hardener, the glue achieved a bending strength of 110 ÷ 120 MPa and flexural modulus of 2.700 ÷ 3.300 MPa. The epoxy glue (LG 385, HG 385) consisted of two parts, which were combined in a 2:1 ratio by volume, according to the product standard’s requirements [45]. 

### 2.3. Fibrous Composites

The glulam wood elements were reinforced using BFRP rods with the diameter of 10 mm, which were glued into the earlier milled grooves with the cross-section of 14 mm × 14 mm. 

The reinforcement in all grooves featured a buffer zone of approx. 2mm and was linked with wood with the use of epoxy glue. The BFRP materials were degreased with the “Aceton” solvent prior to glue application. The composite rod was fixed and pre-stressed to approx. 30 MPa (which corresponds to a 2 mm elongation of the BFRP rod along the beam’s length) on the supports, and then the epoxy glue (LG 385, HG 385) was applied by filling in the square grooves along the beam’s entire length [45]. The flexural moduli and final deformations of the BFRP basalt rods used for reinforcing wooden beams, taken from the manufacturer’s data sheet, were as follows: E = 78 GPa, *Ɛ*_u_ = 39% [45].

The beams were placed in a room with stabilised conditions of relative humidity (65 ± 5%) and temperature (20 ± 2 °C) for at least 7 days to ensure complete glue curing.

## 3. Testing Program

The glulam beams were subjected to bending tests according to PN-EN 408+A1:2012 [59]. The glulam beams had the length of 3650 mm and cross-section with lateral dimensions of 82 mm × 162 mm. They consisted of four sawn wood lamellas with various quality class configurations. The cross-sections of the analysed beams are presented in Figure 2.

The testing featured series of beams with worse mechanical properties using BFRP rods to test the effectiveness of the reinforcement. All the beams were tested in a four-point bending system. The span of support axles amounted to 3000 mm. The configuration of the tested beams and methods of their reinforcement are presented in Figure 2. Each configuration featured testing of three beams in a natural scale to ensure the results’ reliability [45]. 

The beams supported freely on both ends were loaded symmetrically, in two points with concentrated forces. Figure 3 presents the wooden beams’ loading plan during the bending test. The vertical displacement measuring sensors were mounted after placing the beams on the strength testing machine’s supports. Each adopted load level featured a reading of displacements and deformations. Another reading was conducted after 20 min. Then, the testing featured relief to the earlier level and repeated reading, then the samples were loaded to the next force level [45]. This scheme was adopted considering the beam’s actual working and the displacement increment over time. 

The testing featured recording of the loading force and displacement, in the midspan and symmetrically, in relation to the beam’s centre in a 5 h span according to PN-EN 408+A1:2012 [59] (see Figure 5), for particular loading levels with simultaneous measurement of wood deformation and fibrous composites’ deformation using a mechanical extensometer with the measuring base of 203.2 mm (8 inches), according to the distribution presented in Figure 4 and Figure 5 [45].

## 4. Test Results and Analysis

As result of the conducted testing, it was ascertained that the deformation and stress values occurring in the wood’s cross-section were lower for beams reinforced with fibrous composites than for non-reinforced beams. Figure 6, Figure 7, Figure 8 and Figure 9 present examples of charts “*ɛ-F/2—including time during loading*”, “*σ—base no.*” for the Wa-A1 beam. Figure 6 and Figure 7 also present the damage observed during the testing.

Figure 6 shows the dependence of tensile and compressive deformations on loading forces in time measured by extensometer at individual bases along the length of a wooden beam. Tensile deformations included lamellas I and II, while compressive stresses—lamellas III and IV. Bases 1 and 13 were located at the support zones, and base 7—at the center of the beam span. As can be seen, in the middle of the beam span (base 7) of the tensile zone (lamella I) there was a significant increase in tensile deformation of wood caused by the increasing knot cracking at 10 kN load.

Figure 7 shows the dependence of tensile deformations on loading forces in time for all bases along the length of the basalt rod located in the lamella I. One can notice the uniform distribution of tensile deformations in the BFRP rod and the elastic work of the fiber composite despite the weakening in the wood. Only a slight increase in tensile deformations occurred at bases 7 and 8 (beam span center) at 20 kN load due to the adhesive detaching from the rod.

Figure 8 shows the distribution of normal tensile and compressive stresses for loading forces over time along the entire length of a wooden beam at bases 1 and 13 (support zones) and at base 7 (center of span). In the central part (base 7) of the lower part of the beam (lamella I), there is a visible increase in stresses in the wood resulting from a wood defect in the form of knots occurring in this place.

Figure 9 presents the chart of stresses along the BFRP rods’ length. The charts present the impact of the wood structure on the stresses in the BFRP rod. The figure shows the uneven increase in stress in the BFRP rod under the influence of increasing loading forces. Visible significant increase in tensile stress occurring in the middle of the beam span. This was due to the epoxy adhesive peeling off from the basalt rod due to the defect in the lamella I. The BFRP rod effectively absorbed the tensile forces in the damaged bottom of the wooden beam.

In reinforced beams, normal stresses were smaller by approx. 16% in relation to NWa-type non-reinforced beams [45].

The uneven distribution of deformations and normal stresses were caused by the occurrence of wood flaws, such as, among others, knots, spiral grain, burls, cracks, uneven growth ring width [Figure 6 and Figure 8] and local loosening of glue from the rod [Figure 7 and Figure 9]. In non-reinforced beams, the wood flaws substantially affected the destruction form (especially in the beam’s stretched zone), whereas in reinforced beams, it was the fibrous composites that substantially affected the beam’s destruction form. The wood destruction was firstly visible in the compressed zone. The BFRP rod substantially limits the impact of flaws located in the beam’s bottom part on the form of the beam’s destruction. The fibrous composites intercepted the tensile forces and limited the deformations of lower wood fibres [45]. After visible plasticisation of the beam’s compressed zone, the reinforced beams’ destruction progressed through of loosening of glue from the rod [Figure 7].

Figure 10 presents the chart of “*u-F/2—including time during loading*” dependencies of the “Wa” type beams for sensor 2. For medium and lower quality class, in the “loading-relief” testing scheme (Wa-A, Wa-B types), the average values of deflections were lower than the values obtained for the NWa type. For unreinforced beams (type NWa), the highest deflection values were characteristic for the NWa-2 beam (25.46 mm at 15 kN load). Considering the beams reinforced with two BFRP rods (type WaA), the Wa-A3 (21.7 mm) beam showed the highest deflection values at 15 kN load. Beams reinforced with three BFRP rods (type WaB) were characterized by smaller average deflection values than WaA beams. 

The form of destruction of non-reinforced beams was classic. The stretched zone featured cracks related to wood flaws (mainly knots). Because of the low quality of the used wood, medium and lower quality class non-reinforced beams became damaged, as a result of defects or irregularities in the stretched zone. The beams with the reinforcement inserted in the stretched zone were working until a non-linear destruction. The beams’ plasticisation values depended on the quality of wood present in the bottom lamella along with the inserted reinforcement [45]. 

The examples of destruction images varied for specific beams (see Figure 11, Figure 12, Figure 13 and Figure 14). Figure 11 presents the Wa-A1 beam’s destruction in the compressed fibre zone, and then in the stretched fibre zone near the wood flaws (37 kN).

Figure 12 presents the propagation of wood fibre cracks in the Wa-A2 beam near the knots, which were followed by rupturing. The destruction occurred in the location of a hidden flaw (a knot), followed by loosening of epoxy glue from the wood (31 kN).

Figure 13 presents the crushing of the Wa-B1 beam’s compressed zone, and then the destruction of the stretched zone near wood flaws (32 kN).

Figure 14 presents knot cracking that occurred at the load of 25 kN. Then, the cracking of burls near knots and spiral grains started to occur. The stretched zone’s destruction occurred at the load of 41 kN.

The beams’ destruction usually resulted from the non-uniform wood structure, i.e., flaws such as knots, local annual growth ring distortion and fibre distortion near knots, catfaces, bark pockets and resin blisters, non-uniform deviation from the rectilinear layout of wood fibres, spiral grain, etc. The wood-glue-rod connection breakage occurred only after the appearance of substantial wood cracks (especially knots). In reinforced beams, in both the compressed and stretched zones, the neutral axis was located approximately at half height of the beam’s cross-section. The reinforcement of pre-stressed basalt rods was substantially changing the destruction forms of the testing models. The non-reinforced models were clearly destroyed in the beam’s stretched zone, whereas the reinforced beams were firstly destroyed by plasticisation of the beam’s compressed zone, and then destruction of the stretched zone (glue-wood adhesion rupturing). 

Table 3 presents the average maximum moments (*M_max_*) for particular types of beams. 

## 5. Conclusions

Based on the conducted testing of non-uniform glulam beams, reinforced with BFRP rods, the following conclusions were drawn:A pre-stressed BFRP rod placed in the stretched fibre zone has satisfactorily improved the tested beams’ bearing capacity.The glulam wood reinforced with pre-stressed BFRP rods achieved higher bearing capacities in the “loading-relief” testing scheme—bearing capacity of approx. 47% and rigidity of approx. 10.4% were achieved, whereas for Wa-B beams, bearing capacity of approx. 54% and rigidity of approx. 9.8% were attained. It is possible to see that the BFRP basalt rods perfectly compensated the non-uniform wood structure and improved the material’s reliability.In locations of wood defects (knots, spiral grain, cracking, resin blisters, etc.), we observed increase in wood and fibrous composite deformations and stresses.The presence of composite rods halts or limits the propagation of cracking. Furthermore, as seen in u-F/2 (deflection–force) type curves, there is a lower spread of test results because of cracking neutralisation.In lower class elements, the destruction usually occurred in the stretched zone through cracking of wood fibres near the wood flaws, i.e., knots. The pre-stressed BFRP basalt rods improved the interoperation between the cracking knot and the “glue-rod” connection that stiffened it. Then, the beam damage usually started in the compressed zone.

The presented results of laboratory tests confirm the effectiveness of strengthening weakened wood zones, in particular those under tensional stresses, with fiber composites obtained by other researchers, and presented in the available literature on the subject. However, the presented studies cannot be directly compared with those of other authors, because no publications were found regarding similar studies using the method of reinforcing with pre-tensioned basalt rods. These results should be regarded as qualitative rather than quantitative.

## Figures and Tables

**Figure 1 materials-12-03141-f001:**
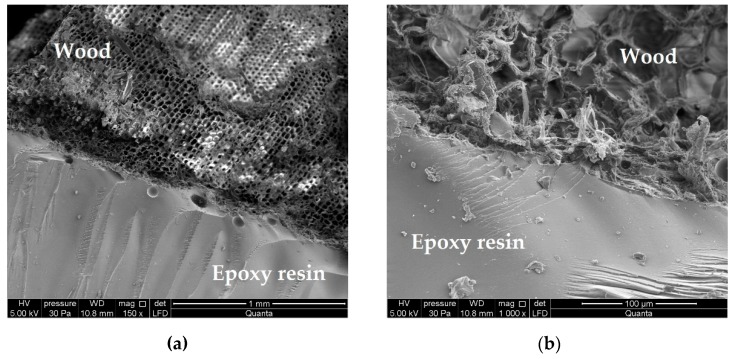
Microstructure of the connection of the LG 385 + HG 385 epoxy resin and *Pinus Sylvestris* L. wood in the beam reinforced with basalt rods, subjected to fuve-year atmospheric impact and biological degradation: (**a**) 150× zoom, (**b**) 1000× zoom.

**Figure 2 materials-12-03141-f002:**
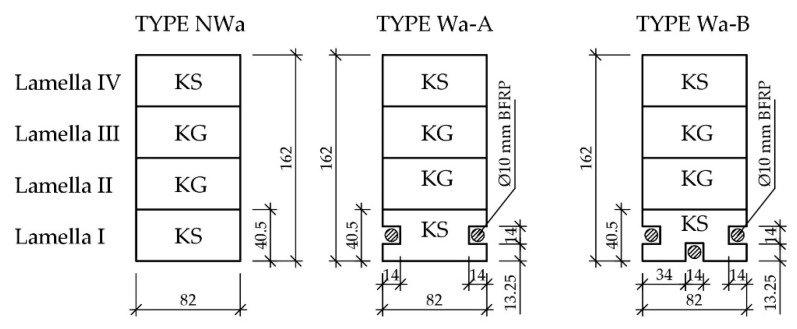
Transverse cross-sections of “Wa” type beams [dimensions in mm, KS—medium quality class, KG—lower quality class].

**Figure 3 materials-12-03141-f003:**
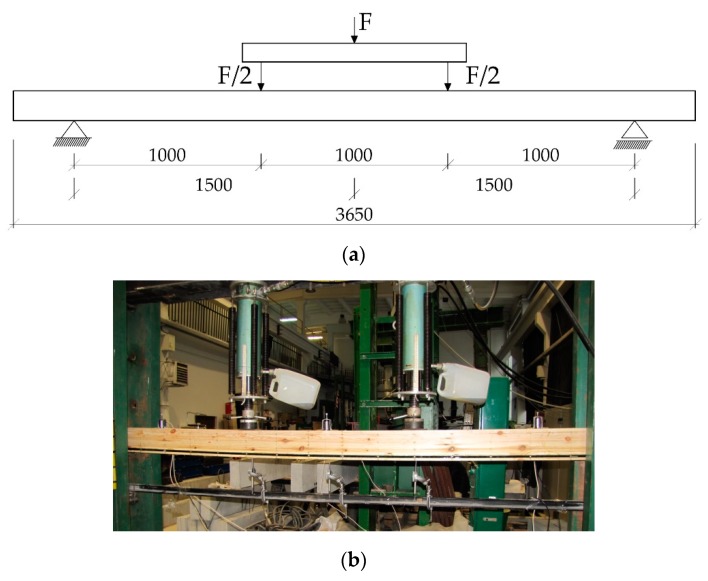
(**a**) Testing station plan: static scheme [dimensions in mm], (**b**) view of the test stand for the Wa-B2 beam.

**Figure 4 materials-12-03141-f004:**
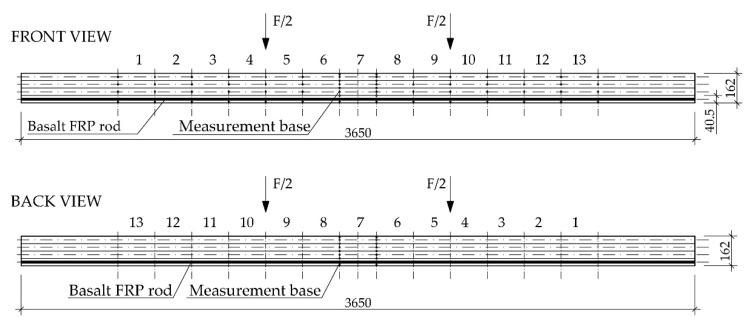
Arrangement of measuring bases in the front and back of the reinforced beams [dimensions in mm].

**Figure 5 materials-12-03141-f005:**
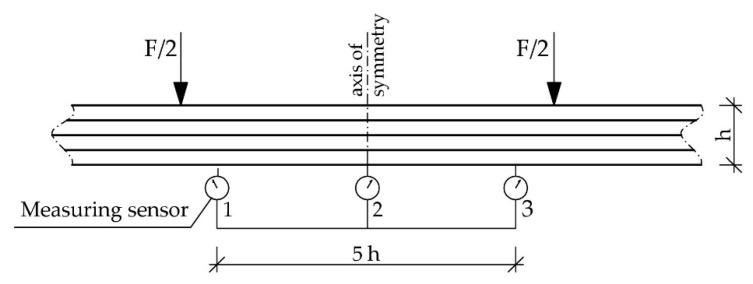
Arrangement of measuring sensors.

**Figure 6 materials-12-03141-f006:**
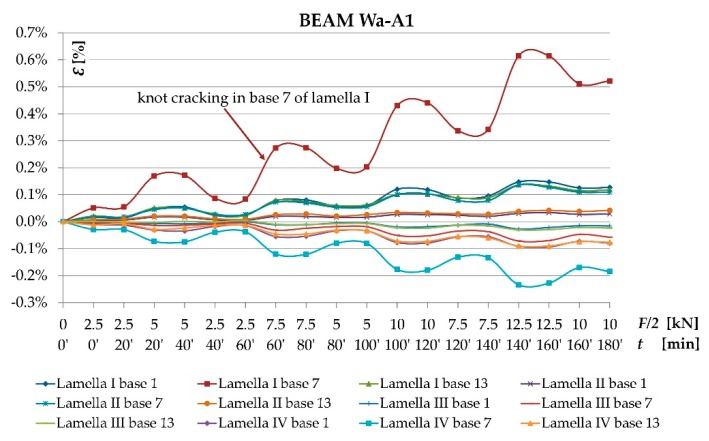
Chart of *Ɛ* [%] and *F/2* [kN]—*including time during loading* dependency for the Wa-A1 reinforced beam.

**Figure 7 materials-12-03141-f007:**
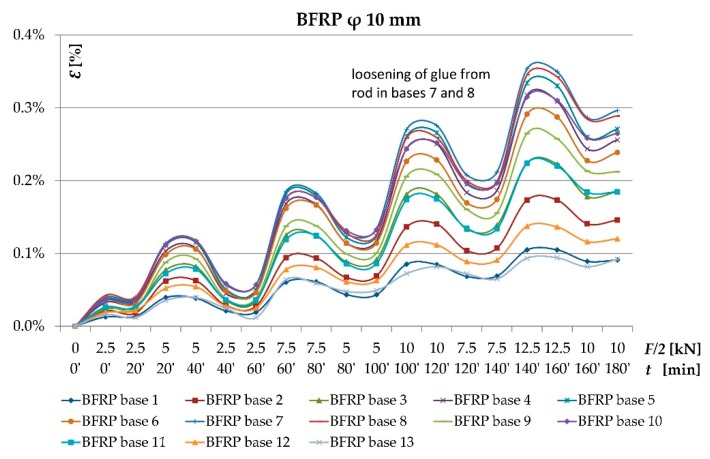
Chart of the BFRP rod’s *Ɛ* [%] and *F/2* [kN]—*including time during loading* dependency for the Wa-A1 reinforced beam.

**Figure 8 materials-12-03141-f008:**
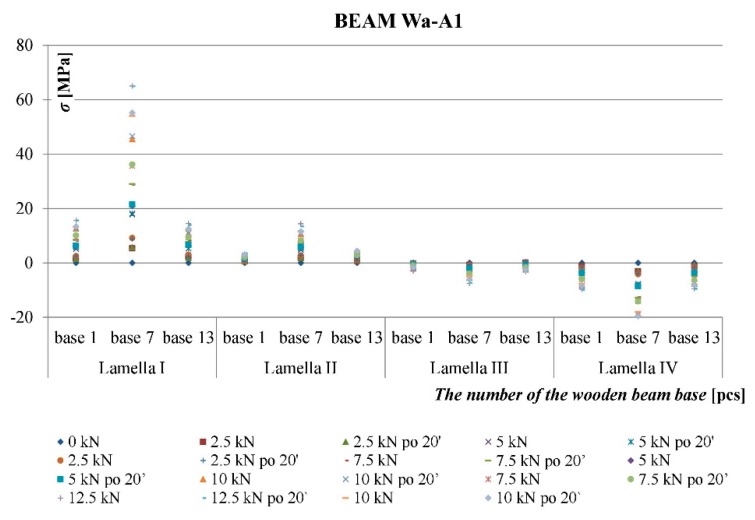
Distribution of normal stresses in the wood *σ* [MPa] along the entire length of the Wa-A1 reinforced beam.

**Figure 9 materials-12-03141-f009:**
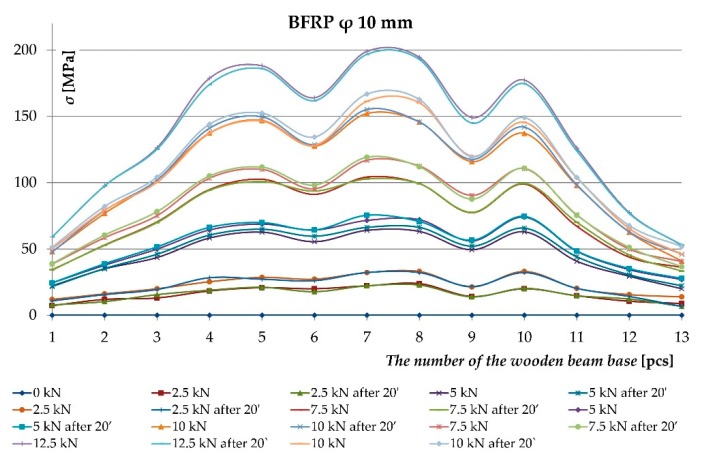
Distribution of normal stresses in the BFRP rod *σ* [MPa] along the entire length of the Wa-A1 reinforced beam.

**Figure 10 materials-12-03141-f010:**
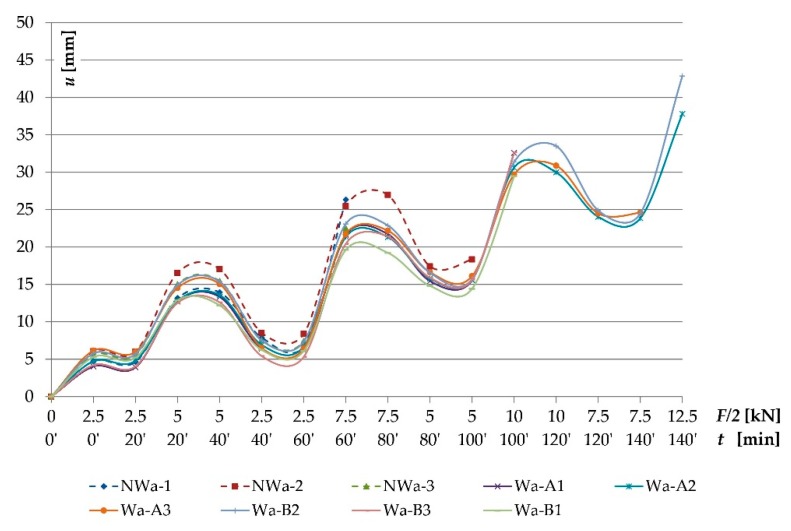
Chart of “*u-F/2—including time during loading*” dependencies of the “Wa” type beams for sensor 2 (beam’s midspan).

**Figure 11 materials-12-03141-f011:**
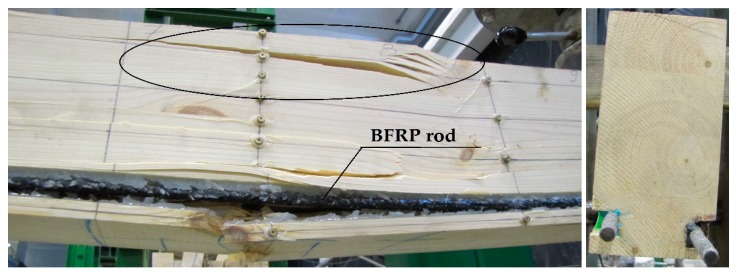
Example image of the Wa-A1 beam destruction in the stretched zone, visible plasticisation of the compressed zone (photograph by Wdowiak).

**Figure 12 materials-12-03141-f012:**
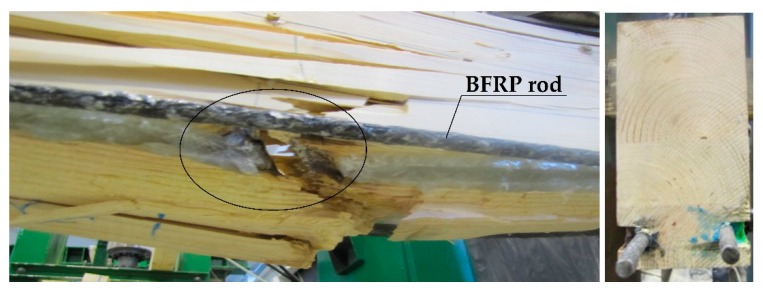
Image of the Wa-A2 beam destruction (photograph by Wdowiak).

**Figure 13 materials-12-03141-f013:**
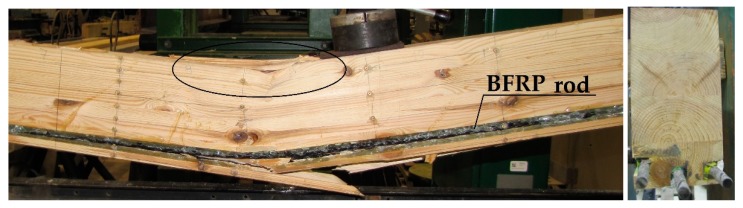
Crushing of the Wa-B1 beam’s compressed zone and the destruction of the stretched zone near wood flaws (photograph by Wdowiak).

**Figure 14 materials-12-03141-f014:**
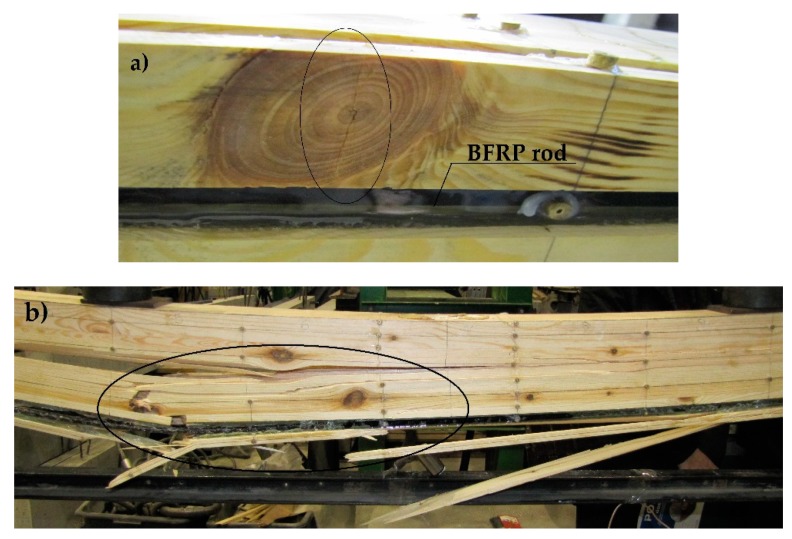
Image of the Wa-B2 beam’s destruction: (**a**) cracking of a deteriorating oval knot, (**b**) cracking of burls near knots and spiral grains (photograph by Wdowiak).

**Table 1 materials-12-03141-t001:** Specification of the types of beams made from medium and lower quality class wood, reinforced with a pre-stressed BFRP rod with a “loading-relief” scheme (*Pinus sylvestris* L.), (including LG—epoxy resin series, HG—hardeners series).

BEAM TYPE	DESCRIPTION
**NWa**	non-reinforced medium and lower quality glulam beams
**Wa-A**	reinforced medium and lower quality glulam beams, BFRP rod reinforcement, pre-stressed, 10 mm diameter, (LG 385 + HG 385), reinforcement ratio 1.18%
**Wa-B**	reinforced medium and lower quality glulam beams, BFRP rod reinforcement, pre-stressed, 10 mm diameter, (LG 385 + HG 385), reinforcement ratio 1.76%

**Table 2 materials-12-03141-t002:** Example of description of structural and geometrical features for selected glulam beams (including USC—general knotty rate, USM—marginal knotty rate).

**NWa-1**	**Front**: healthy oval knot at the edge (lamella I, bases 1, 3, 6, 9, lamella II, base 12, lamella III, base 4), healthy oval knot on the plane (lamella I, bases 5, 8, 13, lamella II, base 3, lamella IV, base 11), degraded knot at the edge (lamella III, base 12, lamella IV, bases 3, 8), deteriorating knot at the edge (lamella IV, base 6), deteriorating knot on the plane (lamella III, base 9), deteriorated knot on the plane (lamella II, bases 9, 13, lamella III, base 7), burls (lamella I, bases 7, 8, lamella III, base 7). **Back**: healthy oval knot on the plane (lamella I, base 13, lamella III, base 13, lamella I, III, IV, base 4 and 5), resin blister on the plane (length 35 mm, depth 1.8 mm, lamella III, base 9), burls (lamella III, base 7).
	Lamella I—USM = 0.41, USC = 0.23, spiral grain 9.4%, growth ring occurrence 4.8 mm (share of late wood 1.3 mm), Lamella II—USM = 0.56, USC = 0.31, spiral grain 10.3%, growth ring occurrence 6.1 mm (share of late wood 0.9 mm), Lamella III—USM = 0.51, USC = 0.29, spiral grain 10.1%, growth ring occurrence 6.4 mm (share of late wood 1.0 mm), Lamella IV—USM = 0.47, USC = 0.24, spiral grain 9.2%, growth ring occurrence 6.3 mm (share of late wood 1.2 mm),
	Density 397.82 kg/m^3^.
**Wa-A1**	**Front**: healthy oval knot at the edge (lamella III, base 7, lamella IV, bases 1, 3, 6), deteriorating oval knot at the edge (lamella III, base 2), deteriorated oval knot on the plane (lamella II, base 4), deteriorated knot at the edge (lamella I, base 7), healthy oval knot on the plane (lamella II, base 8), deteriorating oval knot on the plane (lamella I, base 11, lamella III, base 10, lamella IV, base 11), healthy oval knot on the plane (lamella II, base 13), longitudinal knot on the plane (lamella IV, base 13), burls (lamella IV, base 8). **Back**: healthy oval knot on the plane (lamella I, bases 7, 9, lamella II, bases 8, 9, lamella III, bases 7, 13, lamella IV, bases 1, 3), deteriorating oval knot at the edge (lamella I, bases 1, 4, lamella IV, bases 8, 9), healthy oval knot at the edge (lamella I, base 5, lamella III, bases 2, 5), deteriorated oval knot at the edge (lamella II, base 6, lamella III, base 3, lamella IV, base 13), deteriorating oval knot on the plane (lamella I, base 5, lamella IV, base 13), resin blister on the plane (length 33 mm, depth 1.8 mm, lamella II, base 10, lamella IV, base 12), burls (lamella III, bases 7, 8, lamella IV, base 8).
	Lamella I—USM = 0.44, USC = 0.18, spiral grain 9.4%, growth ring occurrence 6.2 mm (share of late wood 1.2 mm), Lamella II—USM = 0.62, USC = 0.33, spiral grain 10.1%, growth ring occurrence 6.9 mm (share of late wood 0.8 mm), Lamella III—USM = 0.58, USC = 0.32, spiral grain 10.7%, growth ring occurrence 8.4 mm (share of late wood 0.9 mm), Lamella IV—USM = 0.42, USC = 0.21, spiral grain 9.0%, growth ring occurrence 6.3 mm (share of late wood 1.4 mm),
	Density 406.11 kg/m^3^.

**Table 3 materials-12-03141-t003:** Maximum moments (*M_max_*) transferred by the “Wa” type beams (medium and lower quality glulam beams, “loading-relief”).

BEAM NWa	M_max_ [kNm]	BEAM Wa-A	M_max_ [kNm]	BEAM Wa-B	Mmax [kNm]
NWa-1	12.50	Wa-A1	18.50	Wa-B1	16.00
NWa-2	9.50	Wa-A2	15.50	Wa-B2	20.50
NWa-3	12.00	Wa-A3	16.00	Wa-B3	16.00
AVERAGE	11.33	AVERAGE	16.67	AVERAGE	17.50
INCREASE [%]	-	INCREASE [%]	47%	INCREASE [%]	54%
Conf. int. (0.05), kNm	2.71		2.71		4.38
Relative error (0.05), %	23.9%		16.3%		25%
Conf.int. (0.1), kNm	1.75		1.75		2.83
Relative error (0.1), %	15.4%		10.5%		16.2%

The increase in reinforcement degree featured a substantial increase in the destructive force and the maximum deflection slightly decreased.

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
