# Peer review of "Effectiveness of Reinforcing Bent Non-Uniform Pre-Stressed Glulam Beams with Basalt Fibre Reinforced Polymers Rods"

_materials, 2019, doi:10.3390/ma12193141_

Round 1

Reviewer 1 Report

Experimental methodology is well described. At the same time, there is room for improvement in the discussion section: 

1) Very superficial analysis of graphs of Fig. 6-10. It is desirable to deepen it so that the reader has a better understanding of the results obtained in complex experiments and in the conclusions.

If one of graphs is difficult to analyze proper within the article, it is better to opt it out.

2) The percentage effect of basalt rods on non-uniform pre-stressed glulam beams mechanical properties is not clear. The numerical measurement error is not taken into account, but as shown below, it is relatively high and this requires statistical comparisons of variants.

Table 3. Missing calculations

  NWa Wa-A Wa-B
1 12.5 18.5 16.0
2 9.5 15.5 20.5
3 12.0 16.0 16.0
Average, kNm
11.3 16.67 17.5
Conf. int. (0.05), kNm  2.71 2.71 4.38
Relative error (0.05), % 23.9% 16.3% 25%
Conf.int. (0.1), kNm 1.75 1.75 2.83
Relative error (0.1), % 15.4% 10.5% 16.2%

Much of data comes from source nr. 45, including the Table 3. Question: Is this an authors's thesis? That should be explained in the text.  

3) The conclusions must be derived from the text of the article and cannot be transferred from another source.

4) Row 323: 6. Patents? If patents exist, they should be listed there. If not, then point 6 is redundant.

Author Response

Dear Reviewer,

Thank you very much for the review.

We would like to thoroughly answer the questions posed.

1) Very superficial analysis of graphs of Fig. 6-10. It is desirable to deepen it so that the reader has a better understanding of the results obtained in complex experiments and in the conclusions.

If one of graphs is difficult to analyze proper within the article, it is better to opt it out.

The article has been supplemented by the descriptions in the charts shown in Figures 6-10. Detailed descriptions that have been introduced into the article are given below:

Figure 6 shows the dependence of tensile and compressive deformations on loading forces in time measured by extensometer at individual bases along the length of a wooden beam. Tensile deformations included lamellas I and II, while compressive stresses - lamellas III and IV. Bases 1 and 13 were located at support zones, and base 7 – at the center of the beam span. As can be seen, in the middle of the beam span (base 7) of the tensile zone (lamella I) there was a significant increase in tensile deformation of wood caused by the increasing knot cracking at 10 kN load. Figure 7 shows the dependence of tensile deformations on loading forces in time for all bases along the length of the basalt rod located in the lamella I. One can notice the uniform distribution of tensile deformations in the BFRP rod and the elastic work of the fiber composite despite the weakening in the wood. Only a slight increase in tensile deformations occurred at bases 7 and 8 (beam span center) at 20 kN load due to the adhesive detaching from the rod. Figure 8 shows the distribution of normal tensile and compressive stresses for loading forces over time along the entire length of a wooden beam at bases 1 and 13 (support zones) and at base 7 (center of span). In the central part (base 7) of the lower part of the beam (lamella I) there is a visible increase in stresses in the wood resulting from a wood defect in the form of knots occurring in this place. The figure shows the uneven increase in stress in the BFRP rod under the influence of increasing loading forces. Visible significant increase in tensile stress occurring in the middle of the beam span. This was due to the epoxy adhesive peeling off from the basalt rod due to the defect in the lamella I. The BFRP rod effectively absorbed the tensile forces in the damaged bottom of the wooden beam. For unreinforced beams (type NWa), the highest deflection values were characteristic for the NWa-2 beam (25.46 mm at 15 kN load). Considering the beams reinforced with two BFRP rods (type WaA), the Wa-A3 (21.7 mm) beam showed the highest deflection values at 15 kN load. Beams reinforced with three BFRP rods (type WaB) were characterized by smaller average deflection values than WaA beams.

2) The percentage effect of basalt rods on non-uniform pre-stressed glulam beams mechanical properties is not clear. The numerical measurement error is not taken into account, but as shown below, it is relatively high and this requires statistical comparisons of variants.

Due to the large-scale tests and the small number of trials, no detailed statistical studies were carried out in the article, it was limited to providing average values. Table 3 has been supplemented with the reviewer's suggestion. The research indicates a positive effect of strengthening and ensuring the safety of beam elements. The authors plan further research to better identify the issue.

Much of data comes from source nr. 45, including the Table 3. Question: Is this an authors's thesis? That should be explained in the text.

The following study is based on the results of own research during the preparation of the doctoral dissertation of one of the authors, marked in article [45]. The above explanation is also included in the article.

3) The conclusions must be derived from the text of the article and cannot be transferred from another source.

Good point! It was an editorial mistake. The conclusions contained in the work are the conclusions of the authors of this work. The recall was deleted in the article.

4) Row 323: 6. Patents? If patents exist, they should be listed there. If not, then point 6 is redundant.

Good point! We apologize for the obvious mistake.

Thank you very much for your valuable comments.

We tried to put all the comments in the article.

Thank you very much

Reviewer 2 Report

The paper shows a very interesting topic. It is well structured and mostly comprehensible.

It would be nice if some pictures of the experimental setup could be added. In addition, the pictures which are listed are not self-explanatory. If there would be added labels of interesting areas resp. important details this would improve understanding a lot.
This is especially relevant for figures 1, 11, 12 and 13.

Additionally a diagram with the loading profile in dependency of time would be helpful as well.

The designation of source 19 is incomplete.

Author Response

Dear Reviewer,

Thank you very much for the review.

We would like to thoroughly answer the questions posed.

The paper shows a very interesting topic. It is well structured and mostly comprehensible.

It would be nice if some pictures of the experimental setup could be added. In addition, the pictures which are listed are not self-explanatory. If there would be added labels of interesting areas resp. important details this would improve understanding a lot.
This is especially relevant for figures 1, 11, 12 and 13.

Good point! Drawings and explanations have been added to the text.

Additionally a diagram with the loading profile in dependency of time would be helpful as well.

The changes have been taken into account.

The designation of source 19 is incomplete.

Thank you very much for your valuable insight! The missing part of the title has been completed.

Thank you very much for your valuable comments.

We tried to put all the comments in the article.

Thank you very much.

Reviewer 3 Report

This paper presents the testing designating the impact of structural non-uniformity on the effectiveness of reinforcing bent wooden beams with basalt fibre rods. In my opinion, the article does not contribute anything new to the state of knowledge and also presents certain weaknesses and shortcomings as a scientific article. Certain aspects need to be improved

Introduction.

The Introduction does not reflect the latest research in this field. The references used are obsolete and do not show the latest advances. There is a need for an in-depth and up-to-date review of the latest advances in knowledge. For example:

Morales-Conde, M. J., Rodríguez-Liñán, C., & Rubio-de Hita, P. (2015). Bending and shear reinforcements for timber beams using GFRP plates. Construction and Building Materials96, 461-472. Ponomarev, A. N., & Rassokhin, A. S. (2016). Hybrid wood-polymer composites in civil engineering. Magazine of Civil Engineering, (8). Bakalarz, M. (2019). The flexural capacity of laminated veneer lumber beams strengthened with AFRP and GFRP sheets. Czasopismo Techniczne2019(Volume 2), 85-96. Gand, A., Yeboah, D., Khorami, M., Olubanwo, A., & Lumor, R. (2018). Behaviour of strengthened timber beams using near surface mounted Basalt Fibre Reinforced Polymer (BFRP) rebars. Engineering Solid Mechanics6(4), 341-352. Gómez, E. P., González, M. N., Hosokawa, K., & Cobo, A. (2019). Experimental study of the flexural behavior of timber beams reinforced with different kinds of FRP and metallic fibers. Composite Structures213, 308-316. .....

2. Testing materials

In Table 1, terms are no defined (NWa, Wa-A, Wa-B). Besides, in Table 1, how is defined "medium and lower quality". Which standard is used for this?. Define "LG" and "HG". How many beams are prepared and tested? At least 30 pieces are needed to obtain significant and extrapolable results.

2.1. Wood

Table 2. Please define the terms "USM" and "USC". The scheme of the lamellas must be presented before Table 2 (Figure 2). Define ALL THE BASES of the lamellas and the entire section with a scheme. Provide some image of the beams tested. 

4. Test results

Representation Figures shown must be improved and the axis better defined. No data about the number of the beams tested is shown. What information provides each Figure. Do the Figures 6-10 show mean values of the all beams tested?

5. Conclusions

The conclusions should be to reflect the knowledge input of this research with respect to existing research as well as to be linked to other existing works.

At finish it is recommended that the English language be reviewed by a native speaker.

Author Response

Dear Reviewer,

Thank you very much for the review.

We would like to thoroughly answer the questions posed.

This paper presents the testing designating the impact of structural non-uniformity on the effectiveness of reinforcing bent wooden beams with basalt fibre rods. In my opinion, the article does not contribute anything new to the state of knowledge and also presents certain weaknesses and shortcomings as a scientific article. Certain aspects need to be improved.

Introduction.

The Introduction does not reflect the latest research in this field. The references used are obsolete and do not show the latest advances. There is a need for an in-depth and up-to-date review of the latest advances in knowledge. For example:

Morales-Conde, M. J., Rodríguez-Liñán, C., & Rubio-de Hita, P. (2015). Bending and shear reinforcements for timber beams using GFRP plates. Construction and Building Materials, 96, 461-472. Ponomarev, A. N., & Rassokhin, A. S. (2016). Hybrid wood-polymer composites in civil engineering. Magazine of Civil Engineering, (8). Bakalarz, M. (2019). The flexural capacity of laminated veneer lumber beams strengthened with AFRP and GFRP sheets. Czasopismo Techniczne, 2019(Volume 2), 85-96. Gand, A., Yeboah, D., Khorami, M., Olubanwo, A., & Lumor, R. (2018). Behaviour of strengthened timber beams using near surface mounted Basalt Fibre Reinforced Polymer (BFRP) rebars. Engineering Solid Mechanics, 6(4), 341-352. Gómez, E. P., González, M. N., Hosokawa, K., & Cobo, A. (2019). Experimental study of the flexural behavior of timber beams reinforced with different kinds of FRP and metallic fibers. Composite Structures, 213, 308-316. .....

Thank you for your valuable comments! Most of the publications are known to us. The article is mainly limited to recalling publications on research on full-size models related to reinforcements with basalt rods. Articles cited in the review relate to research on small research models and mostly do not apply to basalt rods.

Testing materials

In Table 1, terms are no defined (NWa, Wa-A, Wa-B). Besides, in Table 1, how is defined "medium and lower quality". Which standard is used for this?. Define "LG" and "HG". How many beams are prepared and tested? At least 30 pieces are needed to obtain significant and extrapolable results.

The materials were sorted visually according to PN-D-94021:2013-10 (KW - exclusive class, KS - medium quality class, KG - lower quality class, and designated for “rejection”). Due to the high costs, three full-size research models were made in each series.

2.1. Wood

Table 2. Please define the terms "USM" and "USC". The scheme of the lamellas must be presented before Table 2 (Figure 2). Define ALL THE BASES of the lamellas and the entire section with a scheme. Provide some image of the beams tested.

According to PN-D-94021: 2013-10, USC - general knotty ratio, USM - marginal knotty rate. Figure 4 defines the location of all bases on individual slats. Appropriate annotation was introduced in the text of the article. Due to the layout of the article, drawing 2 was not moved before table 2, before table 2 a reference to this drawing was introduced in the text.

Test results

Representation Figures shown must be improved and the axis better defined. No data about the number of the beams tested is shown. What information provides each Figure. Do the Figures 6-10 show mean values of the all beams tested?

Three models in the series were made in the research. Figures 6-9 present graphs in relation to one selected Wa-A1 beam reflecting the effect of heterogeneity of the structure. In contrast, Figure 10 shows the charts for all tested models.

Conclusions

The conclusions should be to reflect the knowledge input of this research with respect to existing research as well as to be linked to other existing works.

The presented results of laboratory tests confirm the effectiveness of strengthening weakened wood zones, in particular stretched ones, with fiber composites obtained by other researchers, and presented in the available literature on the subject. However, the presented studies cannot be directly compared with those of other authors, because no publications were found regarding similar studies using the method of reinforcing with pre-tensioned basalt rods. These results should be regarded as qualitative rather than quantitative.

At finish it is recommended that the English language be reviewed by a native speaker.

Thank you for your remark! Anyway, the article has been independently checked and the linguistic correctness was confirmed by a  proofreading service.

Thank you very much for your valuable comments.

We tried to put all the comments in the article.

Thank you very much.

Round 2

Reviewer 3 Report

The authors have improved the paper with the changes suggested. However, the introduction and the conclusions must be revised in order to include the comments of the last review. After these minor changes the paper can be accepted for this reviewer. 

Author Response

Dear Reviewer,

Thank you very much for valuable comments.

The authors have improved the paper with the changes suggested. However, the introduction and the conclusions must be revised in order to include the comments of the last review. After these minor changes the paper can be accepted for this reviewer. 

We introduced literature references to the article and added additional conclusions that result from our own research and cited papers.

Thank you very much.
